

# Accurate humidity probe for persistent aviation-contrail conditions

Christoph Dyroff[1], Michael Moore[1], Bruce C. Daube[1], Scott C. Herndon[1]

[1]Aerodyne Research, 45 Manning Road, Billerica MA 01821, USA

*Correspondence to*: C. Dyroff (cdyroff@aerodyne.com)

**Abstract.** We present a state-of-the-art humidity probe for *in situ* airborne water vapor measurements tailored to the humidity range relevant for the formation of persistent aviation contrail cirrus clouds. Our probe is based on tunable infrared laser direct absorption spectroscopy (TILDAS). We scan a laser in wavelength across an isolated water ($H_2O$) absorption line near 7205 cm$^{-1}$ (1.39 μm) to obtain the $H_2O$ volume mixing ratio in sample air at 1 Hz in real time. Our novel optical design features a combination of single-mode optical fibers and a short-path absorption cell with integrated focusing optics and detector to

avoid any absorption path outside of the sample volume. A parallel fiber path and detector capture the laser power profile during the wavelength scan for spectral-baseline normalization. We have built and tested two prototypes and tested them side-by-side in the laboratory against a humidity standard as well as against each other. Without any calibration, we found agreement against the humidity standard of better than 98% for the relevant $H_2O$ range below 200 ppm. The agreement between both instruments when operated in series measuring the same sample air was 99.7%. The stability of both instruments was quantified

to be ±1% (1 σ) during a 5-day long period where both prototypes operated without any temperature control. We show that our probes can measure $H_2O$ with linear response over 4 orders of magnitude.



## 1 Introduction

Persistent aviation contrail cirrus clouds are increasingly recognized as a dominant contributor to the non-$CO_2$ climate impacts
of aviation (Kärcher 2018, Quaas et al., 2021, Voigt et al., 2021, Spangenberg et al., 2013). These clouds form when water
vapor from aircraft exhaust condenses and freezes in the cold upper troposphere, particularly under ice-supersaturated
conditions. If ambient atmospheric conditions are favorable, such as sufficiently low temperatures and high humidity with
respect to ice (RHi), these contrails can persist and evolve into extensive cirrus cloud fields, contributing significantly to the
Earth's radiative budget through their ability to trap outgoing longwave radiation (Lee et al., 2021; Bock and Burkhardt, 2016).
The effective radiative forcing (ERF) from contrail cirrus is now estimated to exceed that of aviation $CO_2$ emissions over the
same time period. According to the updated assessment by Lee et al. (2021), the ERF of contrail cirrus is approximately
57.4 mW/m$^2$, compared to 34.3 mW/m$^2$ from aviation-induced $CO_2$, highlighting their substantial contribution to aviation's
climate impact. Consequently, the need for accurate understanding and mitigation of contrail cirrus has become increasingly
urgent.
A central factor in the formation and persistence of contrail cirrus is atmospheric humidity (Appleman, 1953). Specifically,
the local RHi determines whether the ambient conditions support contrail formation and whether these ice crystals will
sublimate or persist (Schumann, 1996; Minnis et al., 2004). Supersaturation with respect to ice is a prerequisite for persistent
contrails. Small variations in humidity at cruising altitudes dramatically affect their lifecycle. Therefore, spatially and
temporally resolved humidity measurements are essential for accurate modelling and prediction.
*In situ* measurements of atmospheric humidity and temperature have proven critical to improving our understanding of contrail
microphysics and their radiative effects. Campaigns such as MOZAIC/IAGOS (Marenco et al., 1998; Petzold et al., 2015) and
CARIBIC (Brenninkmeijer et al., 2007) have provided large-scale, long-term observational datasets using commercial aircraft
platforms, allowing the evaluation of supersaturated regions and the frequency of contrail formation. These datasets are also
foundational to the development of contrail-avoidance strategies, where flight trajectories are dynamically adjusted to avoid
regions with high RHi (Teoh et al., 2022). The application of such strategies in flight planning has shown promising results in
simulations, with the potential to reduce contrail-related climate forcing significantly (Matthes et al., 2020).
Research-grade airborne humidity sensors have been demonstrated and further enhanced the capability of the scientific
community. These include frost-point hygrometers (Hall et al. 2016, Vömel et al. 2016), Lyman-α fluorescence sensors (Meyer
et al. 2015, Sitnikov et al. 2007), mass spectrometers (Kaufmann et al. 2016), and tunable diode laser absorption spectrometers
(Zondlo et al. 2010, Dyroff et al. 2015, Buchholz et al. 2017, Sarkozy et al. 2020, Graf et al. 2021). These instruments have
been used for precise water vapor measurements, but they are one-off instruments for dedicated research campaigns.
The only commercially available and/or flight certified instruments are the WVSS-II (Vance at al. 2015) formerly offered by
Spectra Sensors and now by FLYHT (https://flyht.com/weather-sensors/wvss-ii/), and the IAGOS core humidity sensor
(Helten et al, 1998, see also https://www.iagos.org/iagos-core-instruments/h2o/) developed by Forschungszentrum Jülich
GmbH in cooperation with enviscope GmbH and is manufactured by enviscope GmbH under licence agreement.



Here we describe a novel laser-based humidity probe specifically designed for, but not limited to, measurements of humidity levels relevant to aviation contrail cirrus. Our design is simple, robust, and fully autonomous. In this paper we demonstrate the performance of two prototypes in a laboratory setting.

## 2 Instrument design

### 2.1 Optics

Our humidity probe is based on direct laser absorption spectroscopy using an all-fiber based optical system. We use a fiber-coupled distributed feedback (DFB) diode laser (AERODIODE, France) at a wavelength of 1.39 µm (7205 cm$^{-1}$) to probe an isolated absorption line of $H_2O$. We use a 50:50 fiber-based beam splitter (Thorlabs, TW1430R5A1) to produce two fiber paths of similar optical power from the primary fiber; the sample path and the baseline path (see Figure 1). For the sample path we

use a single-mode fiber pigtail with glass ferrule termination (Thorlabs, SMPF0115-APC). We use a graded-index (GRIN) lens (Thorlabs, GRIN2313A) to create a slightly focused laser. Both the fiber ferrule and the GRIN lens are permanently glued to a stainless-steel holder using UV-curing adhesive (Thorlabs, NOA61). The small gap between the ferrule and the GRIN lens is bridged with the same glue, which is transparent at the working wavelength and has a similar refractive index to the fiber ferrule and GRIN lens. The alignment on the centered optical axis is performed in a jig with the GRIN lens held in a 3-axis

translation stage while the glue is not cured. The beam is aligned through a pinhole target onto a detector at 300 mm. Both the pinhole and the detector are located precisely centered on the optical axis. Once alignment is achieved, the glue is immediately cured using ultraviolet (UV) light.

The GRIN lens holder has a tapered locating feature, and the sample cell has an equivalent tapered bore, similar to tapered tool holders, e.g. in a milling machine. This allows the GRIN lens holder to be placed centered on the cell's optical axis without

any optical adjusters. An O-ring on the tapered face creates a vacuum seal and allows replacement of the GRIN lens assembly should it be required.

The opposite side of the sample cell holds the sample detector (Thorlabs, PDAPC4). The detector is also integrated into the sample cell. To do so, we first bond an anti-reflection coated wedged window (Thorlabs, PS814-C) to the angled (1º) end face of the cell using NOA61. This window forms the gas seal of our single-pass sample cell with internal volume of 72 cm$^3$. We

then align and bond the sample detector onto the outward face of the wedged window using NOA61. The position is defined by two 3D-printed detector holders (not shown in Figure 2). The cell face and wedge angles are chosen such that all reflections from the intermediate surfaces are reflected away from the optical axis or do not re-enter the sample cell entirely. This effectively eliminates optical interference within the sample cell, which could otherwise cause temperature-driven instrument drift. The sample cell has two gas ports for the sample gas in and out flow. A thermistor is threaded into the stainless-steel cell

body.

The cell design with the GRIN lens holder and the integrated sample detector effectively eliminates all optical pathlength outside of the sample cell that could otherwise be subject to generating absorption signal not due to the sample volume. This





is very important for our humidity probe, as the sample will often contain humidity two orders of magnitude lower than the air surrounding the probe, e.g. an aircraft cabin. We thus eliminate any unknown zero offset in our humidity measurements.

The second fiber path created by the fiber Y-splitter is connected to a single-mode fiber pigtail with glass ferrule termination (Thorlabs, SMPF0115-APC). The glass ferrule is directly bonded to the window of the baseline detector (Thorlabs, PDAPC4). We use this detector to measure the optical power profile of the laser diode as it is scanned in wavelength via injection current. The resulting baseline spectrum does not contain any signal due to absorption of $H_2O$. We are using this baseline spectrum to continuously normalize the sample spectrum in our spectroscopic fit model.

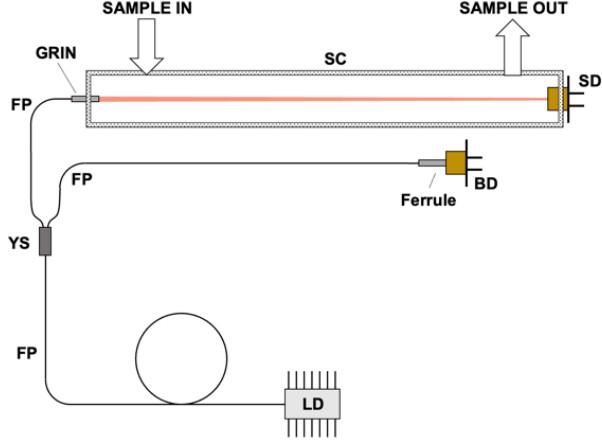


**Figure 1: Schematic of the humidity probe. LD: laser diode; FP: fiber pigtail; YS: fiber Y-splitter; SC: sample cell; SD: sample detector; BD: baseline detector.**

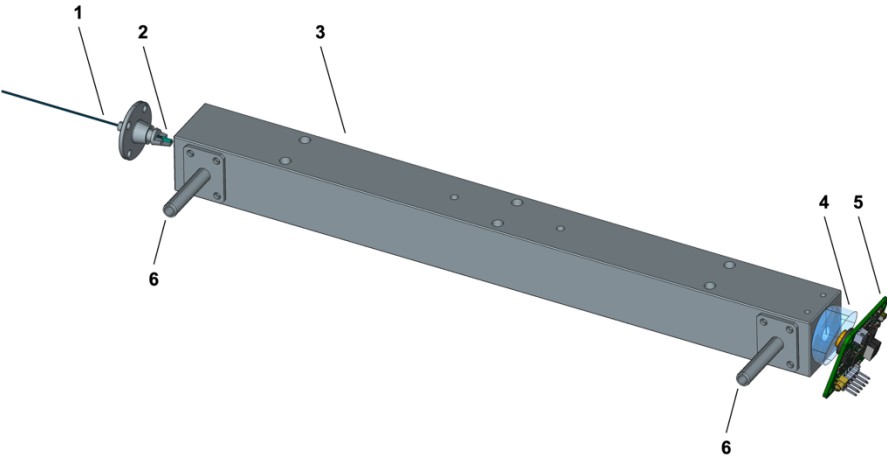

**Figure 2: CAD rendering of the sample cell. Not all components shown. GRIN lens holder not in final position. 1: fiber pigtail; 2: GRIN lens; 3: cell body; 4: wedged window; 5: detector; 6: gas ports.**



## 2.2 Electronics

Two prototypes were built and tested with the same optical system, but with different electronics packages. For prototype 1 we used a data acquisition system equivalent to what we use in the larger commercial Aerodyne Research TILDAS instruments.

A personal computer (PC) operated the in-house TDLWintel software package. The software communicated with a set of National Instruments (NI) data acquisition cards. The laser was driven using a state-of-the-art laser driver (Wavelength Electronics QCL-125) with very low current noise. Upon pre-averaging of the spectra, they were fit in TDLWintel at 1 Hz in real time to produce $H_2O$ mixing ratios using the measured sample-gas temperature and pressure. For prototype 2 we used a different electronics system (RedWave Labs, Universal Platform for Spectroscopic Instruments). While similar in principle, it

combines a low power PC, up to 3 laser drivers, and a data acquisition system into a very compact unit. We have developed software that operates the Universal platform with one laser. Spectra were pre-averaged on the unit's field programmable gate array (FPGA) before being transferred to the PC for processing and storage. The lasers and detectors of both prototypes are the same model and vendor.

## 2.3 Spectroscopy

The laser wavelength was rapidly scanned by linearly ramping the laser injection current in time from below threshold to 50 mA (70 mA for prototype 2) at a frequency of 2666 Hz (334 Hz for prototype 2) as shown in Figure 3. The final 10% of the time of each laser scan, the laser was operated below threshold to record the detector dark signal. The spectra were continuously digitized and combined to a one-second average spectrum before being transferred to the PC. For prototype 1 the sample and baseline spectra were recorded interleaved. Every minute, 50 s of 1 Hz sample spectra were followed by one 10 s

average baseline spectrum. To record both spectra with only one analog-to-digital (ADC) converter available we used a signal multiplexer to switch between the sample and baseline detector. For prototype 2 we were able to acquire both sample and baseline detector signals simultaneously using two ADCs at the cost of a lower number of spectra averaged in 1 s. With prototype 2 we recorded spectra on the device and then fit them offline using the same fitting engine as embedded in TDLWintel, though the fit setup was adjusted for the different laser and its scan.

In both systems, the wavelength scale is projected on the time base of the instrument by a laser-dependent non-linear tuning rate function (Figure 3 lower panel). This function was empirically measured for each laser using the resonant fringe pattern of a 25 cm long glass rod (Edmund Optics, Stock #84-531) with uncoated plane-parallel end faces. The maxima and minima of the fringe pattern are equally spaced in wavelength, and we were thus able to infer the relative wavelength tuning of the laser, while the $H_2O$ transition wavelength obtained from the HITRAN database (Gordon et al. 2022) provided the anchor

point in the spectrum for the absolute wavelength scale.





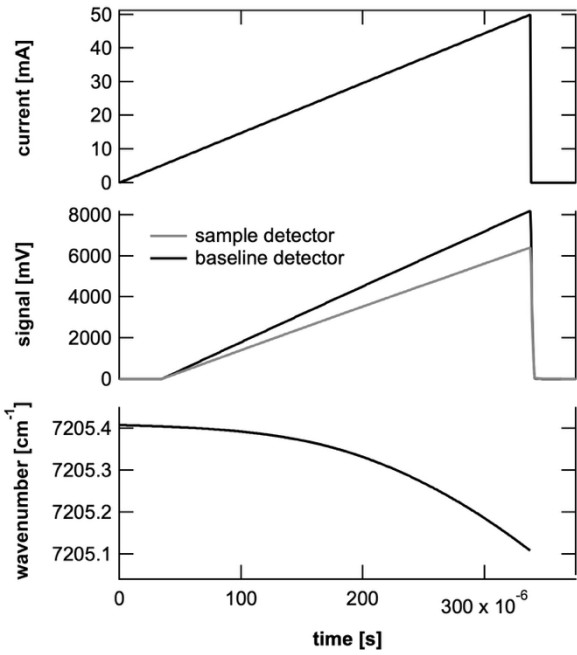

**Figure 3: Laser scan parameters for prototype 1. The prototype 2 laser was scanned slower due to a different hardware configuration. For the low H₂O concentrations discussed here, the H₂O absorption feature cannot be resolved by eye in the sample detector spectrum (grey trace). For a transmission spectrum see Figure 4.**

Figure 4 shows the transmission spectrum of $H_2O$ recorded at 1 Hz by prototype 1 at three different concentrations: 6.5 ppm, 13 ppm, and 26 ppm. The grey dots are the measured baseline-normalized spectra, and the black lines are the fit. The signal to noise is excellent with the 1 Hz noise around 100 ppb in the contrail-relevant humidity range <200 ppmv as shown for both prototypes in Figure 5. The slightly higher noise of prototype 2 is related to the slower scan rate. In both cases the noise is low enough to not significantly affect the overall measurement uncertainty.



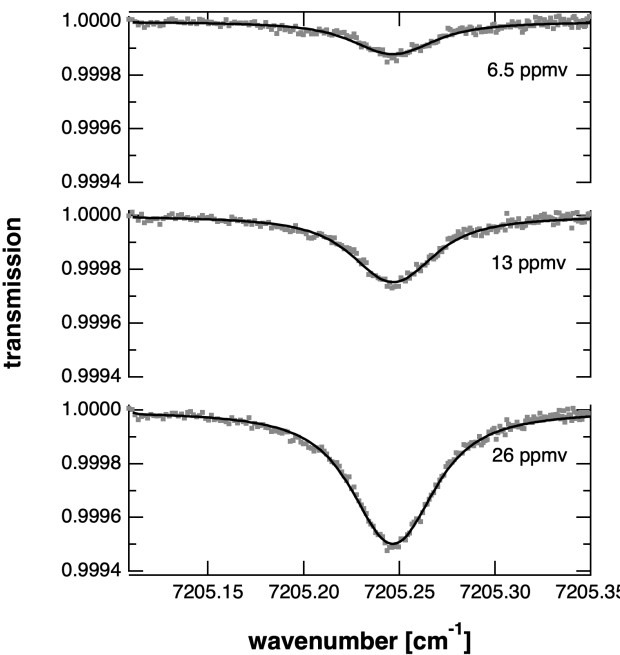


**Figure 4: Measured 1-second average transmission spectra of H₂O at different concentrations. The grey dots are the measurement, and the black lines are the fit.**

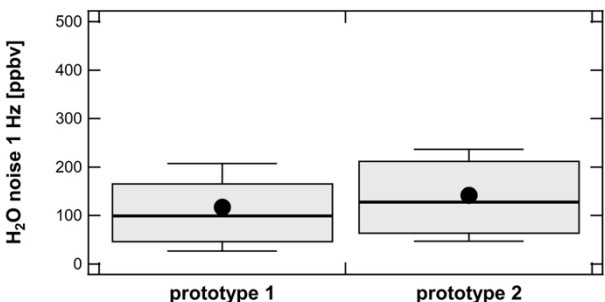

**Figure 5: H₂O noise of both prototypes at 1 Hz measurement frequency defined as** $ADEV = \left(1/2\,(H2O_{i+1} - H2O_i)^2\right)^{1/2}$**.**

**3 Instrument characterizations**

**3.1 Test setup**

For the performance evaluation of our prototype instruments, we have set up the humidity-generation system shown in Figure 6. We used a flow of around 1.5 standard liters per minute (slpm) of ultra-zero air (UZA) with $H_2O < 3$ ppmv from a compressed cylinder as feedstock. The UZA was further dried by passing through a Nafion dryer followed by a trap filled with

molecular sieve at room temperature. The flow through the sample cell (10) and the pressure of the sample gas (11) were set




to 1 slpm and 213 hPa (160 Torr), respectively, using the two manual flow restrictors (5) and (12). A pump (13) generated the required vacuum downstream of the sample cell. The remaining UZA flow was discarded via an overblow port (4).

We used a dew-point generator (6, Licor LI-610) to generate a flow of around 1.5 slpm of saturated humid air set manually by a flow restrictor (7). Of this flow, a small flow of 0 to 10 standard cubic centimeters per minute (sccm) could be directed into

the low-pressure UZA flow via a flow controller (9). The remaining flow was discarded via an overblow port (8). With this system we were able to generate a gas flow with humidity ranging from (near) zero parts per million by volume (ppmv) to around 125 ppmv.

The bubbler of the DPG was exposed to ambient pressure, which was changing with the local weather pattern. This resulted in variations of the saturated absolute humidity generated by the DPG (relative humidity remained at 100%). To this end we

have calculated the expected absolute humidity provided by the DPG taking the ambient pressure into account. We used the Buck formula to calculate the saturated $H_2O$ partial pressure $p_{H2O}$ (in hPa) from the DPG based on the DPG temperature setpoint $T$ (in ºC):

$$p_{H2O} = 6.1121 \cdot exp\left(\frac{18.678-T}{234.5} \cdot \frac{T}{257.14+T}\right). \tag{1}$$

We then calculated the generated $H_2O$ in ppmv using the local atmospheric pressure $p_{atm}$ (in hPa):

$$H_2O = \frac{p_{H2O}}{p_{atm}} \cdot 1E6. \tag{2}$$

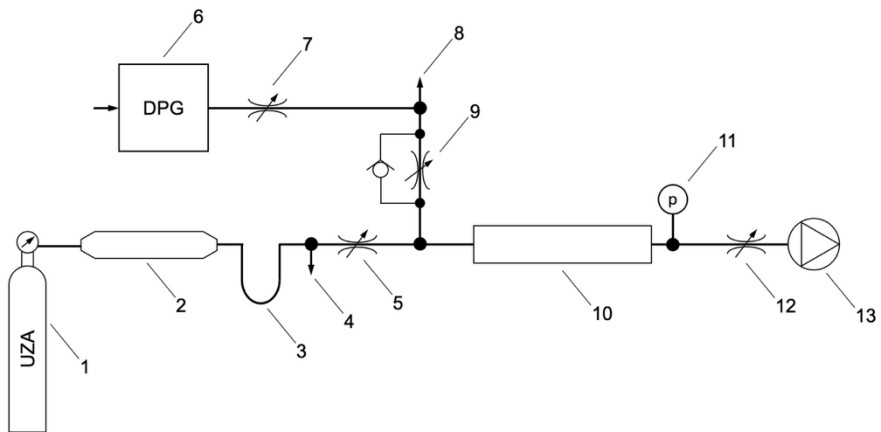

**Figure 6: Schematic of our laboratory humidity-generation system. (1) UZA tank; (2) Nafion dryer; (3) desiccant trap; (4) overblow port; (5) flow restrictor; (6) dew-point generator; (7) flow restrictor; (8) overblow port; (9) flow controller; (10) sample cell; (12) flow restrictor; (13) vacuum pump.**

The saturated humidity from the DPG set to T=10 ºC was typically around 12000 ppmv depending on the atmospheric pressure (Eq. 1 and 2). With a dilution of up to 10 sccm into 1 slpm (100x) the maximum humidity of our generated humidified UZA was around 120 ppmv. The minimum humidity was assumed 0 ppmv, but it is possible that up to 1 ppmv remained after passing the UZA through the Nafion dryer and desiccant. With this setup, our laboratory testing was based around ramping





the generated humidity between dry (nominally 0 ppmv) and around 120 ppmv. Each ramp contained 9 humidity levels with
quasi-randomized order that lasted one hour in total. We remained at every humidity level for 5 minutes and at the final zero-
air level for 20 minutes. An example time series is shown in Figure 7. Unfortunately, our DPG failed shortly after this
measurement, and it remained the only hour of data where both prototypes measured the accurately generated humidity
simultaneously. We show in a later section how well both prototypes agree with each other.

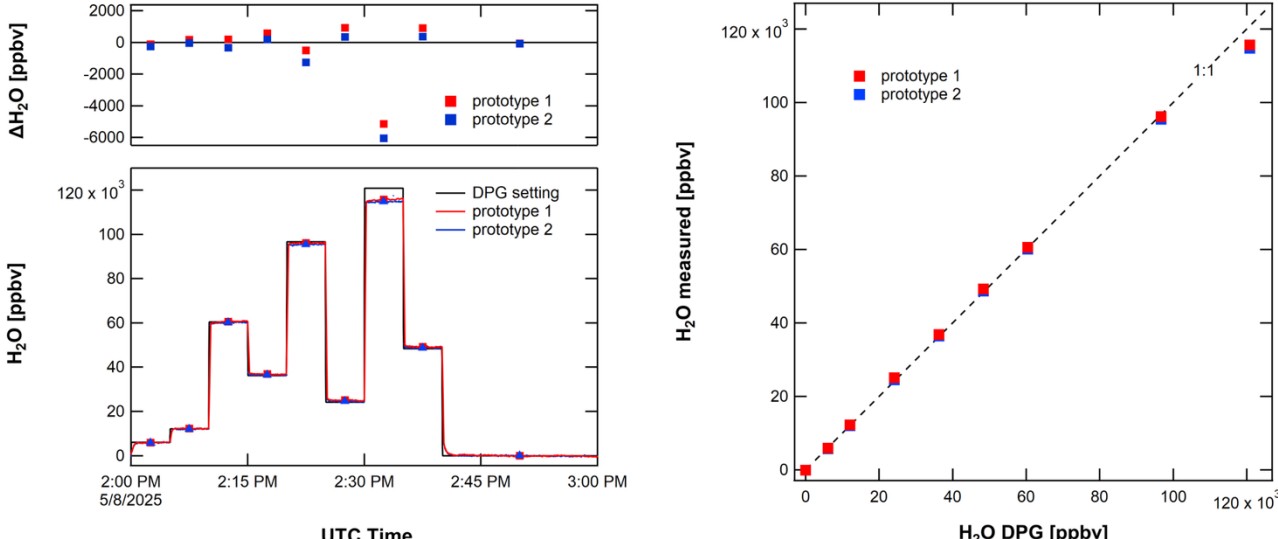

**Figure 7: (left) Humidity ramp as described in text. We note that the flow controller at the highest setting likely did not provide the
expected flow of 10 sccm and thus both prototypes reported lower humidity than expected as shown by difference of measurement
to expectation ($\Delta H_2O$). (right) Measured $H_2O$ of both prototypes against the expected $H_2O$ from the DPG system showing excellent
linearity.**

## 3.2 Accuracy

We have quantified the accuracy of our humidity probe (prototype 1) by performing a 50-hour long measurement of known
humidity in a range of 0 ppmv to around 120 ppmv. The known humidity was delivered to the humidity probes by the humidity-
generation system shown in Figure 6. Each hour of the measurement consisted of 9 humidity levels, 8 levels of 5-minute
duration each followed by a 20-minute period of zero air. The 1-second humidity measurements of each level were averaged
for the last 4.5 minutes of each non-zero level to one average value per level. The final level was zero air, and it was measured
for 20 minutes and averaged for 19.5 minutes. We then performed linear fits of the individual humidity ramps and derived the
slope (measured vs. generated $H_2O$), zero offset, and $r^2$. The fit parameters were unconstrained. Figure 8 depicts the humidity
time series in the top panel, followed by the slope, the zero offset, and $r^2$.





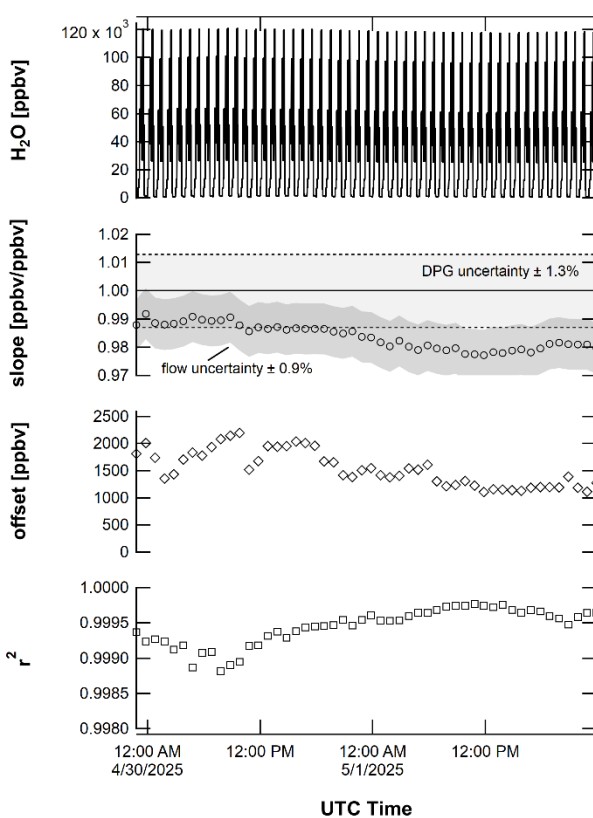

**Figure 8: (top) Time series of a 50-hour long measurement of prototype 1. Each hour was fit with a linear function and the lower**
**panels show the individual slopes, offsets, and goodness of fit ($r^2$). The shaded areas of the slope panel show the uncertainty of the**
**DPG humidity of ±1.3% and the flow uncertainty of ±0.9%.**

On average we found an agreement of the prototype 1 instrument with the expected humidity from the DPG of 0.984±0.004 (slope in Figure 8), i.e. the measured humidity was 1.6% lower than expected. We also found that the zero air appeared to have around 1 ppmv to 2 ppmv of residual humidity, independently of our efforts to dry it further. See the prototype intercomparison below for evidence of this claim. In the future we consider using a liquid-nitrogen cold trap to freeze out the residual humidity to get to a lower zero humidity. The linearity of the individual fits was excellent as indicated by the high $r^2$.

The measured humidity always agreed within the combined uncertainties of the DPG (±1.3%) and the dilution flow controller (±0.9%) as indicated by the overlap of the shaded areas in the slope panel of Figure 8. This is an excellent result considering the prototype has not been calibrated prior to the measurements yet accurately reproduced the generated humidity.

## 3.3 Prototype instrument intercomparison

We have performed a direct comparison of our two prototype instruments by operating them is series gas flow during a period of 5 days. During this time, prototype 1 operated continuously, and prototype 2 operated with extended pauses during which other tests and tasks where performed. In total, both instruments operated for 63 hours simultaneously. Unfortunately, our





DPG became unreliable, and we changed the setup of Figure 6 such that we added laboratory air with variable humidity to the UZA via a flow controller in the same series of steps. While this setup does not provide a reference humidity, it is well suited to compare the two prototype instruments. A short section of the data taken in this experiment is shown in Figure 9. As previously we calculated average $H_2O$ values for every step of the dilution ramp, i.e. 9 average values per hour (not shown in Figure 9).

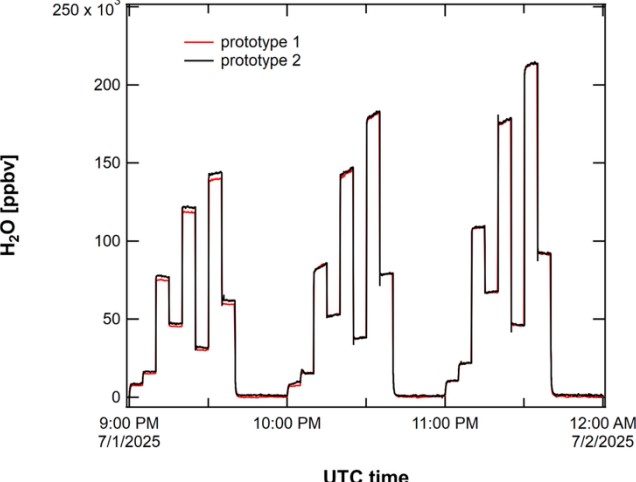

**Figure 9: Section of data of both prototype systems operating in gas-flow series.**

In a first step of our analysis, we performed a direct comparison of the average $H_2O$ measurements of the two prototypes as shown in Figure 10. A linear fit was performed that gave an excellent average agreement between the two sensors of 0.997 ppbv/ppbv with a very low zero offset of 109 ppbv.

We quantified the agreement between the two sensors further by calculating the point-by-point difference. Figure 11 shows the histogram of this difference for all 568 average data points spanning a $H_2O$ range of 0 ppmv to 213 ppmv. The average difference was -61 ppbv, and the standard deviation was 1492 ppbv. The box plot of the difference provides a visual representation of the data distribution, where the box represents the 25% and 75% quantiles, and the whisker lines show the standard deviation. The thick vertical line represents the median (-23 ppbv) For 50% of the N=568 averaged data points the two systems agreed to within ±1000 ppbv (Q25 to Q75), and for 68% to within ±1500 ppbv (±1σ).

We note that neither sensor has been calibrated prior or after the experiments, and both sensors were not temperature controlled in any way and exposed to temperature changes of up to 8ºC/day. This level of accuracy at low mixing ratios enables reliable determination of RHi thresholds critical for predicting whether contrails will persist or evaporate.





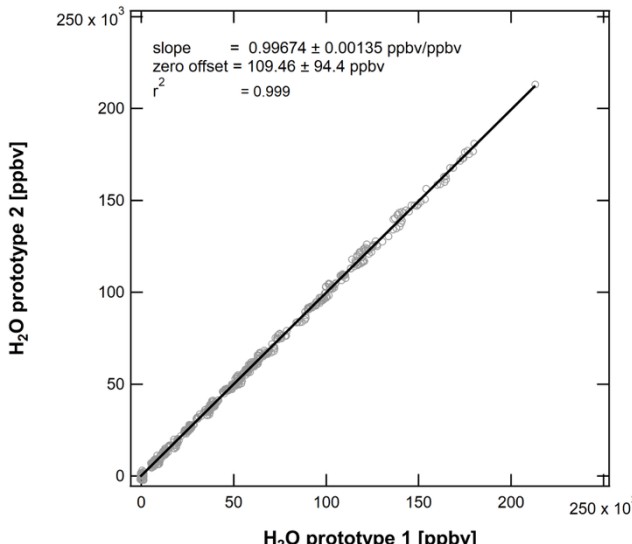

**Figure 10: Direct prototype system intercomparison. The average agreement was 99.7% with a very low zero offset of 109 ppbv.**

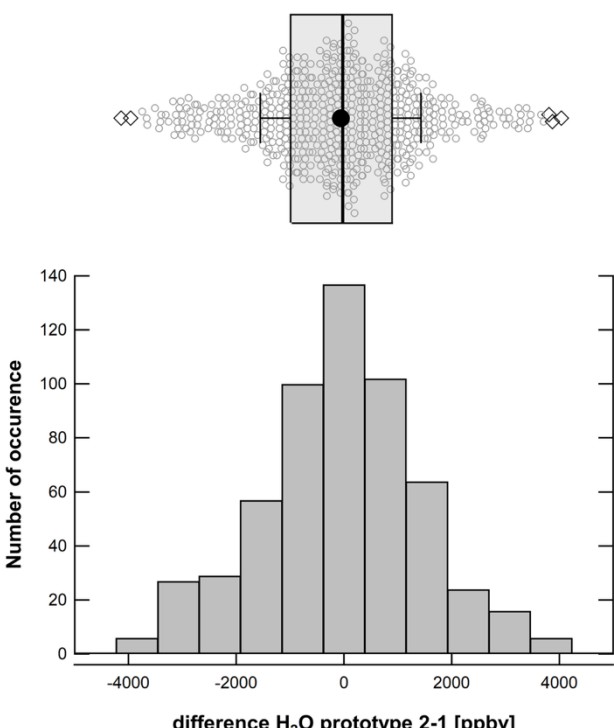


**Figure 11: (bottom) Histogram and box plot of the difference (prototype 2 – prototype 1) of averaged $H_2O$ measurements. (top) Box plot of the difference data indicating median (thick vertical line), average (solid marker), 25% and 75% quantiles (box), vertical lines (±σ).**





### 3.4 Stability

We have quantified the prototype instrument stability using the data obtained during the 5-day period discussed in the previous section. To do so, we have performed linear fits of the 9 measured $H_2O$ average data of every hour and derived the slope, the offset, and the goodness of the fit ($r^2$). These metrics are shown in Figure 12. The initial 12 hours of the test resulted in an agreement of 0.96 to 0.98. After the initial 12 hours we have implemented an improved laser-scan setup to prototype 2, which resulted in a slope between the two instruments closer to 1 (top panel of Figure 12). For the remainder of the test the scan setup

was not changed, and we achieved an agreement (slope) of $1.004 \pm 0.011$ ppbv/ppbv (avg $\pm$ sdev) as indicated by the thick horizontal line and the shaded area between the dashed lines. During the same time the zero offset was $-57 \pm 1079$ ppbv. These results are in excellent agreement with our assessment of the accuracy of our prototype 1 using the DPG absolute humidity scale (Figure 8), where the slope changed by around 1% during a 50-hour long test. We assume that the instrument drift of both systems has no common-mode term, and thus stability within 2% between the two independent instruments was expected.

The demonstrated stability over multi-day operation without temperature control supports integration into long-duration flight missions and routine operations without onboard recalibration.

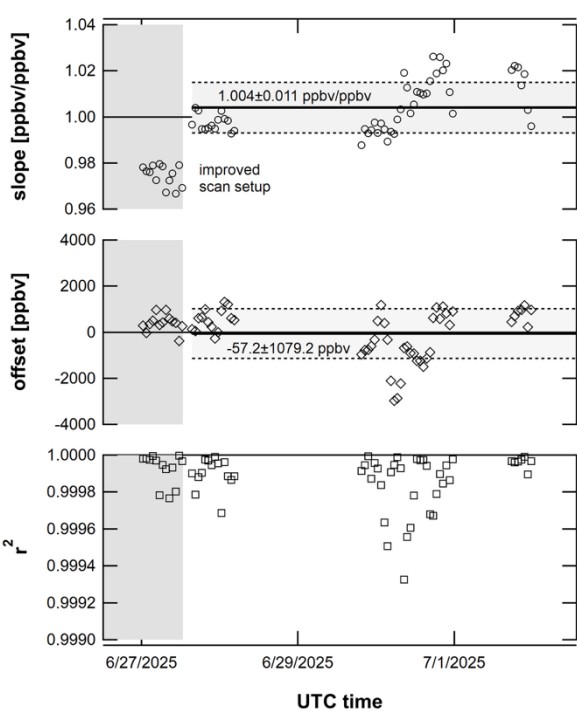

**Figure 12: Direct prototype system intercomparison showing hourly (top) slope, (middle) zero offset, and (bottom) goodness of linear fit. Average and standard deviations of slope and offset for the improved laser-scan setup are given in the figure.**



## 3.5 Extended humidity range

We have tested the linearity over a humidity range spanning four orders of magnitude. While the focus for our humidity probe is the humidity relevant for contrail cirrus formation ($H_2O < 150$ ppmv) our sensor also supports much higher humidity measurements. For the high humidity range, we have measured the undiluted output of the DPG. We have changed the temperature setpoint of the DPG between 2ºC and 20ºC to generate $H_2O$ between around 6900 ppmv and 23000 ppmv. We measured at each DPG temperature for around 5 to 10 minutes and calculated the average value for each level. The results are shown in Figure 13 together with $H_2O$ measurements of the low humidity range obtained by diluting the DPG output into UZA. For prototype 1, we found that the measured $H_2O$ showed non-linearity above around 7000 ppmv. We were familiar with such an effect from our commercial mid-IR TILDAS instruments, where optical-power dependent saturation of the detectors causes non-linearity of the retrieved gas concentration. We have thus experimented with attenuating the optical power incident on both the sample and the baseline detectors by adding either a 50/50 or a 25/75 fiber-based y-splitter (YS) upstream of the existing 50/50 y-splitter of our fiber assembly (YS in Figure 1). Using either the 50% or the 25% output port of the additional YS generated a 50% or 75% attenuation of the optical power. We have then repeated the undiluted $H_2O$ measurements with prototype 1 and found that the attenuation mitigated the non-linear response of our sensor. With 75% attenuation we were able to extend the linear range to around 15000 ppmv. The measurements above 15000 ppmv showed remaining non-linear response, but this can in principle be included in a calibration. Prototype 2 showed much better linearity as shown by the blue data markers in Figure 13 (inset). It almost matched the linearity of prototype 1 at 75% attenuation. The absolute power levels on the detectors of both prototypes were similar, where prototype 2 had around 25% less optical power on both detectors. This may explain some of the better linearity of prototype 2 but not all in our opinion. Both detector pairs (sample and baseline) of the two prototypes were purchased at different times of the development phase and likely come from different manufacturing batches, which may explain the differences in their response.





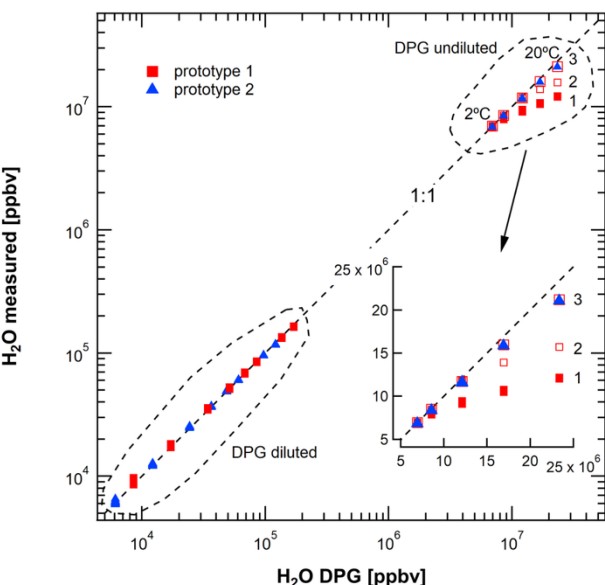

**Figure 13: Measured H$_2$O by prototype 1 (red) and prototype 2 (blue). The optical power on both detectors of prototype 1 was 1: 100%, 2: 50%, and 3: 25%, that of prototype 2 was 100%. The low humidity data of prototype 1 and prototype 2 were recorded with the DPG set to 15ºC and 10ºC, respectively. The inset shows the high humidity range between 5000 and 25000 ppmv with linear axes.**

## 4 Discussion

In this paper we have demonstrated a novel humidity probe for the detection of conditions favoring the formation of persistent aviation contrail cirrus clouds. Our probe is explicitly simple in design. Without calibration the two prototypes built and tested showed excellent agreement against a humidity standard of better than 98% for the relevant H$_2$O range below 200 ppm. The agreement between both instruments when operated in series measuring the same sample air was 99.7%. The stability of both instruments was quantified to be $\pm 1\%$ (1 $\sigma$) during a 5-day long period where both prototypes operated without any temperature control. Furthermore, we show that our probes can measure H$_2$O with linear response over 4 orders of magnitude.

Achieving the high level of accuracy of each prototype required careful setup of the spectroscopic fit, including the laser-dependent tuning rate, the baseline characteristics as well as position and width of the fit window. Our prototypes were operated exclusively at 213 hPa (160 Torr), though the technology is not limited to this pressure. We chose it as it reflects (approximately) the atmospheric pressure at altitudes where aviation contrails are occurring. In an airborne deployment one could (i) control the pressure to a fixed value, or (ii) let the pressure float with the altitude-dependent atmospheric pressure. While the latter approach may be favorable as it would allow to operate the instrument without a vacuum pump, we have not performed the required test matrix to quantify the accuracy in this scenario.

The prototype drift was very low at $\pm 1\%$ (1 $\sigma$) during a 5-day long period. This is a key result and paved the way for the envisaged unattended deployment of our sensors on commercial or research aircraft. The combination of accuracy, stability,



and operational simplicity demonstrated here directly supports the development of automated contrail-avoidance decision-support tools. By providing continuous, reliable humidity measurements in the critical low-ppmv range, our probe enables in-flight detection of ice-supersaturated regions, informing real-time flight path adjustments

In addition to the excellent accuracy at contrail-relevant humidity, our instrument measured humidity spanning 4 orders of magnitude. We identified a non-linear response of our prototype instruments which we attributed to optical-power dependent saturation effects of the detectors. We presented mitigation strategies by (i) lowering the optical power incident on the detectors, and (ii) calibration for linearization at high humidity. Future work will focus on airborne certification, extended flight testing across seasonal and geographic regimes, and integration with meteorological data streams for real-time contrail-

avoidance applications.

*Data availability.* The data shown in this paper are available upon request.

*Author contributions.* CD and SH wrote the manuscript. SH led the SBIR project and wrote the new software package. CD
and BD developed the optical system. MM performed electronics design. SH, CD, MM, and BD conducted the experiments. CD and SH performed data reduction and analysis.

*Acknowledgements.* We thank Dr. Richard Moore at NASA Langley Research Center for the opportunity to participate in the 2023 EcoDemonstratior flight campaign with an early prototype of the described humidity probe. Our final design greatly
benefited from our flight results.

*Financial Support.* This work was funded by NASA via SBIR grant number 80NSSC23CA066.

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
