# Peer review of "Accurate humidity probe for persistent aviation-contrail conditions"

_EGUsphere, 2025_

## Referee Comment (RC1)

**Review of "Accurate humidity probe for persistent aviation-contrail conditions" by Dyroff et al. (2025)**

**Overview**

Persistent contrail cirrus has become a main concern in the aviation industry and scientific community for its radiatively warming impact on the climate. Although the formation mechanism and conditions of persistent contrails are well understood, sensitive and accurate humidity measurements at aircraft cruising altitudes remain the key restriction to improve numerical weather prediction models for robust forecast of persistent contrail conditions.

This paper presents a newly developed humidity probe based on tunable infrared laser direct absorption spectroscopy (TILDAS) that can measure water vapour ( $H_2O$ ) mixing ratios in the range of persistent contrail formation conditions, even down to a few parts per million by volume, with high accuracy, low noise and offset. Without pre-calibration, both prototypes 1 and 2 implementing slightly different electronic systems, data acquisition and analysis software show excellent agreement between each other and with the reference dew-point generator. Besides accuracy, the stability of the two humidity probes was tested in series over 5-day operation, proving the instrument robust. Attenuating optical power improves significantly the linear response of the humidity probes in high  $H_2O$  volume mixing ratios.

Overall, the manuscript is well-written and describes in detail the set-up, characterisation, and test of the new humidity probes. The paper is high relevant, and it fits very well with the scientific scope of AMT. I still have some comments that I like to see addressed before publishing this paper.

**General comments:**

- 1. The author describes the TILDAS humidity probes are tailored to measure humidity promoting contrail formation and are suitable for commercial and research aircraft deployment. The prototypes seem handy to measure low  $H_2O$  mixing ratios in laboratory settings. All the testing described in the paper were done in well controlled lab environment. However, as far as I am concerned, the testing is not complete to demonstrate the reliability of the prototypes for in-situ measurements aboard aircraft, despite the multiple-day operation.
  - 1) The temperature change in the lab was up to 8°C/day (L221). What impact it have on the selected line strength, the accuracy and stability of the probes? Pogány et al. (2015) stated that even the temperature variation during the day in the lab could have a significant effect on the absorption line strength. I would like to see a figure in Sect 3 to show whether the spectra were affected by the temperature change.

- 2) What are the author's views on real atmospheric measurements on aircraft with strong vibration, large pressure variation, weigh larger temperature difference ~70°C between the surface and upper troposphere (than the tested 8°C/day). Did the author perform sensitivity tests on the effect of low pressure and temperature on the spectra, the accuracy and stability of the probes, like in Buchholz et al. (2014), evaluating the instrument in simulated low pressure and temperature environment?
- 3) How is the attenuation of optical power implemented so that the humidity probe can keep its linearity when transitioning from conditions with low H2O values to the ones with higher values? Or sacrifice the measurements in the lower atmosphere?
- 4) Have the authors learned any of these during the EcoDemonstrate flight campaign? It might worth being shared in the discussion.
- 2. It is good that the authors kept the introduction short and focused on the instrumental set-up, characterisation and tests, given it is a technical paper. However, the motivation for developing a new humidity probe might seem a bit weak. The authors listed a range of available precise in situ hygrometers and pointed out they were designed for research campaigns. WVSS-II and IAGOS ICH, as successful aircraft-based  $H_2O$  instrumentation, which are also simple, robust, and autonomous, were mentioned. But are there any limitations in the WVSS-II and IAGOS ICH making the developing of a new humidity probe so pressing and beneficial to the persistent aviation topic?
- 3. Some more information in the instrument design section should be added, e.g., dimension, weight, sample volume, optical path length, and power consumption of the TILDAS humidity probes.
- 4. To support the contrail-avoidance decision tools, the humidity probe should deliver robust and accurate measurements with minimal maintenance and long-term stability. For example, IAGOS ICH sensors are taken back to the lab for calibration after  $\sim$ 500 flight hours to ensure data quality (Neis et al., 2015). We can already see a slight decrease in the slope of prototype 1 in the 50-h period. And in Figure 12, the prototype 2 had to adjust its laser scan to achieve better agreement with the prototype 1. Sometimes, the slope, offset and  $r^2$  still fall out of the uncertainty range despite mostly good agreement. Can the authors explain how to achieve long-term stable and accurate measurements without re-calibration while deploying the probe for long-time routine operations? Any online monitoring and self-adjustment available?
- 5. The design of the TILDAS humidity probe make it preferably installed in the pressurized cabin of an aircraft, like the WVSS-II. Can the authors comment on the ambient temperature measurement that should be used to convert  $H_2O$  mixing ratio to local  $RH_{ice}$ ? Relying on the temperature data from the aircraft itself? As  $RH_{ice}$  governs the fate of persistent contrail cirrus, how large is the uncertainty then in  $RH_{ice}$ ?
- 6. The author did not discuss the response time of the TILDAS humidity probes. Based on Fig. 5 and 7, they seem to respond fast to the change of  $H_2O$  mixing ratios, even below 20

ppmv despite almost negligible crawling effect when switching between zero-air and  $H_2O$  measurements. The IAGOS sensor increases its response time from a few seconds at 273 K to a few minutes below 233 K (Neis et al., 2015). Therefore, it is also of high interest and relevance to see if the response time of the TILDAS increases under cruising conditions.

7. In the discussion section, the authors reclaimed the novelty design of the TILDAS humidity probe and its good accuracy and stability. However, I am not convinced so far by the deployment of such a probe on an aircraft for autonomous measurements for supporting contrail avoidance strategies. The measuring technique employing the absorption of H2O in the near-infrared range has been well established in airborne hygrometers, such as JPL laser hygrometer (May et al., 1998), SHARC (Kaufmann et al., 2018), HAI (Buchholz et al., 2014, 2017), SEALDH-II (Buchholz et al., 2018), which can measure down to a few parts per million with low offsets and uncertainties. Interests and potential exist to have some of the prototypes adapted for routine H2O observations for improving contrail forecasts. Furthermore, WVSS-II using the same technique has been long in operation in the (T)AMDAR network. I suggest the authors extend the discussion when discussing the performance of the prototypes by make cross comparisons with similar instruments in the aspects of technique details, measurement capabilities, airborne deployment feasibility so that the readers can easily follow the novelty, simplicity and reliability of the presented humidity probe. Thus, the suitability of the instrument for measuring in aviation contrail environment presents itself to the readers.

**Specific comments:**

L32 "Supersaturation with respect to ice is a prerequisite for persistent contrails": In addition to ice supersaturation, contrail cirrus may also be persistent in slightly ice subsaturated regions depending on the sizes of ice particles according to Li et al. (2023)

L33 "Small variation in humidity at cruising altitudes dramatically affect their lifecycle" -> "... their life cycle by controlling ice crystal formation, growth and dissipation (Unterstrasser and Gierens, 2010)."

L33 "spatially and temporally resolved humidity measurements" is vague. It should explicitly be high spatial and temporal resolution measurements.

L38: References after contrail formation: Petzold et al. 2020, Li et al. 2023, Gierens et al., 2020

L79: The thermistor type? What is its uncertainty? Only one thermistor is inserted in the cell body. Is it located in the milled of the cell body?

Figure 2: I think the dimensions of the measurement cell worth being noted in the figure.

L113: The data recording interval of sample detector and baseline detector is quite clear. The authors also performed dark signal check. How is this reflected in the data acquisition and analysis? Or is this just recorded for inspection?

Figure 5: The abbreviation ADEV for Allan deviations needs to be explained while it is not note elsewhere.

Figure 6: Element 11 missing in the caption.

Figure 7 (bottom left and right), 8 (top), 9, 10 and 13: I find the units of H2O mixing ratio in "ppbv" are unnecessary because the lowest H2O volume mixing ratio to be detected was a few ppmv.

L201: "is" -> in

L203: "where" -> were

Figure 9: In the first cycle, the prototype 1 obviously measured slightly lower values than the prototype 2 at each step above about 50 ppmv, which was not repeated in the second and third cycles. Do the authors have any ideas on the cause of this?

References: 1 Not all DOI are inserted as links. 2 Not all journal names were abbreviated. 3 The style of the references with a long author list should be unified.

References mentioned in the comments:

Buchholz, B., Afchine, A., and Ebert, V.: Rapid, optical measurement of the atmospheric pressure on a fast research aircraft using open-path TDLAS, Atmos. Meas. Tech., 7, 3653–3666, https://doi.org/10.5194/amt-7-3653-2014, 2014.

Pogány A, Wagner S, Werhahn O, Ebert V. Development and Metrological Characterization of a Tunable Diode Laser Absorption Spectroscopy (TDLAS) Spectrometer for Simultaneous Absolute Measurement of Carbon Dioxide and Water Vapor. *Applied Spectroscopy*. 2015;69(2):257-268. doi:10.1366/14-07575

Buchholz, B. and Ebert, V.: Absolute, pressure-dependent validation of a calibration-free, airborne laser hygrometer transfer standard (SEALDH-II) from 5 to 1200 ppmv using a metrological humidity generator, Atmos. Meas. Tech., 11, 459–471, https://doi.org/10.5194/amt-11-459-2018, 2018.

May, R.D. (1998), Open-path, near-infrared tunable diode laser spectrometer for atmospheric measurements of H2O, *J. Geophys. Res.*, *103*, 19161-19172, doi:10.1029/98jd01678.

Kaufmann, S., Voigt, C., Heller, R., Jurkat-Witschas, T., Krämer, M., Rolf, C., Zöger, M., Giez, A., Buchholz, B., Ebert, V., Thornberry, T., and Schumann, U.: Intercomparison of midlatitude tropospheric and lower-stratospheric water vapor measurements and comparison to ECMWF humidity data, Atmos. Chem. Phys., 18, 16729–16745, https://doi.org/10.5194/acp-18-16729-2018, 2018.

Neis, P., Smit, H. G. J., Krämer, M., Spelten, N., and Petzold, A.: Evaluation of the MOZAIC Capacitive Hygrometer during the airborne field study CIRRUS-III, Atmos. Meas. Tech., 8, 1233–1243, https://doi.org/10.5194/amt-8-1233-2015, 2015.

Li, Y., Mahnke, C., Rohs, S., Bundke, U., Spelten, N., Dekoutsidis, G., Groß, S., Voigt, C., Schumann, U., Petzold, A., and Krämer, M.: Upper-tropospheric slightly ice-subsaturated regions: frequency of occurrence and statistical evidence for the appearance of contrail cirrus, Atmos. Chem. Phys., 23, 2251–2271, https://doi.org/10.5194/acp-23-2251-2023, 2023.

Unterstrasser, S. and Gierens, K.: Numerical simulations of contrail-to-cirrus transition – Part 1: An extensive parametric study, Atmos. Chem. Phys., 10, 2017–2036, https://doi.org/10.5194/acp-10-2017-2010, 2010.

Petzold, A., Neis, P., Rütimann, M., Rohs, S., Berkes, F., Smit, H. G. J., Krämer, M., Spelten, N., Spichtinger, P., Nédélec, P., and Wahner, A.: Ice-supersaturated air masses in the northern mid-latitudes from regular in situ observations by passenger aircraft: vertical distribution, seasonality and tropospheric fingerprint, Atmos. Chem. Phys., 20, 8157–8179, https://doi.org/10.5194/acp-20-8157-2020, 2020.

Gierens, K.; Matthes, S.; Rohs, S. How Well Can Persistent Contrails Be Predicted? *Aerospace* 2020, *7*, 169. https://doi.org/10.3390/aerospace7120169

---

## Referee Comment (RC2)

Review of "Accurate humidity probe for persistent aviation-contrail conditions" by Christoph Dyroff et al., MS No.: egusphere-2025-3972

**Summary:**

This manuscript reports on a humidity sensor for the application of persistent aviation contrail conditions (i.e., humidity levels so high that contrails of condensed ice will remain persistent and not evaporate). It presents a novel optical design featuring optical fibers and a short-path absorption path. Two nearly identical prototypes are compared to each other (the only difference being electronics). The two prototypes show good agreement with each other over a large range of water volume mixing ratios.

**Overall:**

Overall, this manuscript is well-written and appropriate for the scope of AMT, presenting a new sensor prototype to measure humidity from aircraft. The title, abstract, presentation, use of language, and references are all good. The novelty of this sensor is both low-noise performance and optical design (optical fibers and a short-path absorption path). One issue is that the manuscript is lacking a connection between the scientific motivation of accurate humidity measurements at aircraft cruising altitudes and performance requirements for parameters including but not limited to: the desired range of water volume mixing ratio, pressures, temperatures, time resolution, accuracy and precision.

Another issue is that the description is not complete enough for subject matter experts to assess this new sensor. The manuscript is missing important details in the description of the instrument design and operation (the flow system), electronics, spectroscopic fitting, data processing, experimental validation, and how the results relate to the performance requirements. If the authors provided more details, then this would be an important contribution to the scientific literature. I recommend that this manuscript should be considered for publication only after substantial revisions to provide more details as addressed in the science comments below.

**General comments:**

1. First, early in the paper, the connection is not described between persistent contrails and the detailed range of water mixing ratios, pressures, temperatures expected. All that is said is that the "relevant H2O range below 200 ppm" is appropriate (page 15, line 274 and

repeated elsewhere). This is missing detail such as: what is the lower limit of H2O mixing ratio expected in contrail-producing regions? What accuracy of measurement is required for this application of predicting persistent contrails? What is the spatial scale in the atmosphere of high humidity / low humidity regions that would drive a requirement of how fast measurements need to be? For instance, are 1-Hz measurements sufficient?

- 2. What are the performance requirements for sufficiently accurate water measurements at aircraft cruising altitudes? Specifically, what is the requirement measurement dynamic range of water volume mixing ratio, pressures, temperatures? What are the required time response, accuracy and precision?
- 3. Given that there are existing commercial instruments (WVSS-II and IAGSO ICH) and scientific-grade TILDAS hygrometers, what is the motivation for developing a new humidity probe? How is this probe novel? This not made very clear, but it is implied in the abstract that the novelty is single-mode optical fiber (and two channels). Could the authors please state why is the novelty important (compared to existing state-of-the-art sensors) for this application of measuring humidity in persistent contrail-forming conditions?
- 4. For the readers, it is important to know hardware details, including material of the optical cell, and what electronics were used.
- 4a. In the lab measurements presented in this manuscript, what pressures and temperatures were used? Were these measured? Pressure and temperature are essential inputs into fitting spectra (as the water mixing ratio is dependent on them).
- 4b. If these prototypes were deployed in an aircraft, how would design change? H would the air be sampled on an aircraft? How would the sensor be packaged? What would be the pressure and temperature in the sample cell? How do gas control, temperature and pressure control affect the measurements?

**Specific Comments:**

- 5. Spectroscopy: page 3, line 57-58: "an isolated absorption line of H2O". It is important to the readers to know specifically which H2O line? What is the wavelength of the isolated line? Can you show a synthetic spectrum of the expected absorbance versus wavenumber (or wavelength) (for the sensor pathlength)? Can you give some rationale for why this particular water absorption line was selected?
- 6. Detail is lacking on optical cell, such as the pathlength, how the detector is mounted. Specifically on page 3, line 65, "The beam is aligned through a pinhole target onto a detector at 300 mm." Is this (300 mm) the optical pathlength?

7. In the Spectroscopy section, pages 4 and 5, can you provide more spectroscopic details, such as:

7a. In lines 88-89: please clarify how the baseline spectrum is used to normalize the sample spectrum. E.g., does this provide the incident/background intensity for the absorption calculation? Is it possible that trapped water in the laser diode or baseline detector could cause the low bias in the accuracy measurements (section 3.2)?

7b. In lines 106-107: what is the motivation for the two different electronics in prototypes 1 and 2? Are there pros and cons of the different implementations?

7c. In line 111, "2666 Hz (334 Hz)": why were these scan rates selected for prototypes 1 and 2?

7d. How many scans are averaged? Are scans fitted in the electronics or in software?

7e. What is the sampling rate?

7f. In line 114: how is 50 seconds of spectral averaging (followed by 10 s of averaging baseline) relevant to airborne measurements? Can the prototypes deliver 1-Hz data at the required accuracy and precision? (see comment #1 above)

7g. In line 103 "Upon pre-averaging of the spectra, they were fit in TDLWintel" and line 118 "we recorded spectra on the device and then fit them offline using the same fitting engine" – can you please say more about how the spectra are fitted? How many spectra are "pre-averaged"?

- 8. Page 5, lines 121-122: "glass rod" please call this an etalon.
- 9. Page 6, lines 130-134: discussion is completely lacking on how the spectroscopic absorption line is fitted. Can you please say more? Are you using the Beer-Lambert Law? How is pressure broadening treated?
- 10. Page 6, line 133-134: "the noise is low enough to not significantly affect the overall measurement uncertainty" this raises several questions:
- 10a. What is the uncertainty of the spectral line strength (and other properties) for the water absorption line selected?
- 10b. If noise does not significantly affect the overall measurement uncertainty, what is the dominant factor contributing to overall measurement uncertainty?
- 10c. Please provide more detail about how the electronics achieve exceptionally low noise? Please quantity the noise on the spectra.

- 11. Page 7, Figure 5 caption: what are the details of this data? Which data went into making this plot? Can you please define the acronym ADEV (please say "Allan Variance").
- 12. Figures 7, 8,10, 13: this manuscript jumps back and forth between ppm and "ppbv times 103". For better clarity, please consistently plot in the same units, ppmv.
- 13. Section 3.5 how would the additional attenuation affect the accuracy and precision of the low-ppm measurement regime? It would also be valuable for the reader to have quantitative information on incident power limits for linear performance.
- 14. Figure 3: Consider adding etalon peaks to subplot to show how the peak locations translate to wavenumber curve
- 15. Figure 7 and Figure 9 show the quick response time of the instrument in lab setting with the test configurations. Could you comment on the cadence / delay times expected when integrated with an aircraft and the associated sampling system?
- 16. Will the sensors still be non-temperature controlled when integrated on aircraft? How will the range of temperatures and rapid fluctuations in temperature affect the instrument hardware and spectroscopic fitting?
- 17. Section 3.5: Please elaborate on how the non-linearities affect the accuracy and the detection limits mentioned.
- 18. Page 15, Line 278-280: Please provide details of all items listed: "spectroscopic fit, including the laserdependent tuning rate, the baseline characteristics as well as position and width of the fit." How do these contribute to the accuracy?

**Editing Comments:**

- 1. Please check grammar. Several commas are missing where needed.
- 2. Consistently use ppmv (not ppm or ppbv) and define ppmv the first time as "parts per million by volume".
- 3. Page 10, line 201: replace "is" with "in".
- 4. References: please cite references in consistent EGU format.

---

## Author Comment (AC1)

We would like to thank both reviewers for the careful reading of the manuscript and the very constructive comments and suggestions, and we hope to have addressed them satisfactorily.

We have provided our answers to the comments and questions below in line in blue color.

**Review of "Accurate humidity probe for persistent aviation-contrail conditions" by Dyroff et al. (2025)**

**Overview**

Persistent contrail cirrus has become a main concern in the aviation industry and scientific community for its radiatively warming impact on the climate. Although the formation mechanism and conditions of persistent contrails are well understood, sensitive and accurate humidity measurements at aircraft cruising altitudes remain the key restriction to improve numerical weather prediction models for robust forecast of persistent contrail conditions.

This paper presents a newly developed humidity probe based on tunable infrared laser direct absorption spectroscopy (TILDAS) that can measure water vapour ($H_2O$) mixing ratios in the range of persistent contrail formation conditions, even down to a few parts per million by volume, with high accuracy, low noise and offset. Without pre-calibration, both prototypes 1 and 2 implementing slightly different electronic systems, data acquisition and analysis software show excellent agreement between each other and with the reference dew-point generator. Besides accuracy, the stability of the two humidity probes was tested in series over 5-day operation, proving the instrument robust. Attenuating optical power improves significantly the linear response of the humidity probes in high $H_2O$ volume mixing ratios.

Overall, the manuscript is well-written and describes in detail the set-up, characterisation, and test of the new humidity probes. The paper is high relevant, and it fits very well with the scientific scope of AMT. I still have some comments that I like to see addressed before publishing this paper.

**General comments:**

1. The author describes the TILDAS humidity probes are tailored to measure humidity promoting contrail formation and are suitable for commercial and research aircraft deployment. The prototypes seem handy to measure low $H_2O$ mixing ratios in laboratory settings. All the testing described in the paper were done in well controlled lab environment. However, as far as I am concerned, the testing is not complete to demonstrate the reliability of the prototypes for in-situ measurements aboard aircraft, despite the multiple-day operation.

1) The temperature change in the lab was up to 8°C/day (L221). What impact it have on the selected line strength, the accuracy and stability of the probes? Pogány et al. (2015)

stated that even the temperature variation during the day in the lab could have a significant effect on the absorption line strength. I would like to see a figure in Sect 3 to show whether the spectra were affected by the temperature change.

The reviewer is certainly correct that the linestrength of the chosen absorption line depends upon the gas temperature via the temperature-dependent population of the lines' ground-state energy level. We have calculated the temperature dependence of the linestrength of the absosption line probed to be 0.004 $K^{-1}$ using the approximation given in Eq. 1 of Gianfrani et al. 2003. This temperature dependence is part of the spectroscopic fitting algorithm and hence temperature variations of the sample gas do not produce changes in the derived mixing ratio.

We have added a Table with spectroscopic parameters of the probed line and their uncertainties stated in the HITRAN database.

2) What are the author's views on real atmospheric measurements on aircraft with strong vibration, large pressure variation, weigh larger temperature difference ~70°C between the surface and upper troposphere (than the tested 8°C/day). Did the author perform sensitivity tests on the effect of low pressure and temperature on the spectra, the accuracy and stability of the probes, like in Buchholz et al. (2014), evaluating the instrument in simulated low pressure and temperature environment?

No tests of the two porotypes beyond those presented in this paper were performed.

We have added Section 5. Outlook and described how the instrument will be operated.

3) How is the attenuation of optical power implemented so that the humidity probe can keep its linearity when transitioning from conditions with low H2O values to the ones with higher values? Or sacrifice the measurements in the lower atmosphere?

Yes, it would be a choice to make before deployment. We do not plan to implement an option to switch mid-operation.

We added this to the outlook section.

4) Have the authors learned any of these during the EcoDemonstrate flight campaign? It might worth being shared in the discussion.

We have added a paragraph in the discussion.

2. It is good that the authors kept the introduction short and focused on the instrumental set-up, characterisation and tests, given it is a technical paper. However, the motivation for developing a new humidity probe might seem a bit weak. The authors listed a range of available precise in situ hygrometers and pointed out they were designed for research campaigns. WVSS-II and IAGOS ICH, as successful aircraft-based $H_2O$ instrumentation, which are also simple, robust, and autonomous, were mentioned. But are there any limitations in the WVSS-II and IAGOS ICH making the developing of a new humidity probe so pressing and beneficial to the persistent aviation topic?

We have added a wider discussion of existing technology and their limitations to the introduction.

3. Some more information in the instrument design section should be added, e.g., dimension, weight, sample volume, optical path length, and power consumption of the TILDAS humidity probes.

Figure 2 has been updated and now shows the cell dimension.

The sample-cell volume of 72 $cm^3$ was given on Page 3, L74.

We have added specifics of the anticipated packaged humidity probe in the Outlook section, which has been added to the manuscript.

4. To support the contrail-avoidance decision tools, the humidity probe should deliver robust and accurate measurements with minimal maintenance and long-term stability. For example, IAGOS ICH sensors are taken back to the lab for calibration after ~500 flight hours to ensure data quality (Neis et al., 2015). We can already see a slight decrease in the slope of prototype 1 in the 50-h period. And in Figure 12, the prototype 2 had to adjust its laser scan to achieve better agreement with the prototype 1. Sometimes, the slope, offset and $r_2$ still fall out of the uncertainty range despite mostly good agreement. Can the authors explain how to achieve long-term stable and accurate measurements without re-calibration while deploying the probe for long-time routine operations? Any online monitoring and self-adjustment available?

We have added a sentence: *This type of adjustment is a routine task in the build process of any Aerodyne laser spectrometer.*

We are grateful to the reviewer to point this detail out. Indeed, we found a weak correlation of the calibration slope and ambient and sample cell pressures. We are convinced this correlation is not a limitation in the spectroscopic retrieval, but rather a limitation of our humidity delivery system.

In our humidity delivery system, we have set the zero-air flow via a manual needle valve. The flow through this valve is critical and thus depends on the upstream pressure. We determined that the calibration slope decreased with higher pressure, which is exactly what one might expect when the zero-air flow increases with higher pressure.

In our analysis we are not correcting for this rather clear correlation and remain with our statement of uncertainty. However, we include a new figure that shows this behavior and explain our finding in the text of Section 3.4 Accuracy.

5. The design of the TILDAS humidity probe make it preferably installed in the pressurized cabin of an aircraft, like the WVSS-II. Can the authors comment on the ambient temperature measurement that should be used to convert $H_2O$ mixing ratio to local $RH_{ice}$? Relying on the temperature data from the aircraft itself? As $RH_{ice}$ governs the fate of persistent contrail cirrus, how large is the uncertainty then in $RH_{ice}$?

We have added a paragraph in the Discussion that puts the uncertainty of our humidity measurements and those of the measured ambient pressure and temperature provided by the aircraft into context.

6. The author did not discuss the response time of the TILDAS humidity probes. Based on Fig. 5 and 7, they seem to respond fast to the change of $H_2O$ mixing ratios, even below 20ppmv despite almost negligible crawling effect when switching between zero-air and $H_2O$ measurements. The IAGOS sensor increases its response time from a few seconds at 273 K to a few minutes below 233 K (Neis et al., 2015). Therefore, it is also of high interest and relevance to see if the response time of the TILDAS increases under cruising conditions.

We have added a new Section 3.2 Response to humidity changes. The response time during the experiments was derived and is indicated in Figure 6.

Unfortunately, we cannot predict response time in an airborne deployment as it depends on the air intake and sample tubing configuration as well as flow rate. If all parameters were the same as in the lab, we predict slightly better response as the sample cell will be heated.

7. In the discussion section, the authors reclaimed the novelty design of the TILDAS humidity probe and its good accuracy and stability. However, I am not convinced so far by the deployment of such a probe on an aircraft for autonomous measurements for supporting contrail avoidance strategies. The measuring technique employing the absorption of $H_2O$ in the near-infrared range has been well established in airborne hygrometers, such as JPL laser hygrometer (May et al., 1998), SHARC (Kaufmann et al., 2018), HAI (Buchholz et al., 2014, 2017), SEALDH-II (Buchholz et al., 2018), which can measure down to a few parts per million with low offsets and uncertainties. Interests and potential exist to have some of the prototypes adapted for routine $H_2O$ observations for improving contrail forecasts. Furthermore, WVSS-II using the same technique has been long in operation in the (T)AMDAR network. I suggest the authors extend the discussion when discussing the performance of the prototypes by make cross comparisons with similar instruments in the aspects of technique details, measurement capabilities, airborne deployment feasibility so that the readers can easily follow the novelty, simplicity and reliability of the presented humidity probe. Thus, the suitability of the instrument for measuring in aviation contrail environment presents itself to the readers.

We have changed the wording from novel to new.

We appreciate the information that some of these instruments have potential to be adapted for routine H2O measurements. As far as we know they have not been made available as commercial products. We have provided references to the instruments mentioned and invite the reader to study their techniques in detail as discussing all the differences and similarities is beyond the scope of this paper.

We have added a paragraph in the introduction that puts published uncertainties of various instruments from the AquaVIT-4 intercomparison into context.

In the Discussion we have added a paragraph that brings the uncertainty of our humidity measurements into context with uncertainties of atmospheric temperature and pressure measurements. This is relevant as contrail formation depends on relative humidity which in turn depends on p and T. We also added a paragraph that compares uncertainties of other instrument to reference standards to provide context for our results.

**Specific comments:**

L32 "Supersaturation with respect to ice is a prerequisite for persistent contrails": In addition to ice supersaturation, contrail cirrus may also be persistent in slightly ice subsaturated regions depending on the sizes of ice particles according to Li et al. (2023)

We thank the reviewer for pointing out this publication.

L33 "Small variation in humidity at cruising altitudes dramatically affect their lifecycle" -> "… their life cycle by controlling ice crystal formation, growth and dissipation (Unterstrasser and Gierens, 2010)."

Reference included.

L33 "spatially and temporally resolved humidity measurements" is vague. It should explicitly be high spatial and temporal resolution measurements.

This is corrected.

L38: References after contrail formation: Petzold et al. 2020, Li et al. 2023, Gierens et al., 2020

These references are now included.

L79: The thermistor type? What is its uncertainty? Only one thermistor is inserted in the cell body. Is it located in the milled of the cell body?

The thermistor location in the middle of the sample cell body is now indicated in Figure 2. The thermistor type is now specified in Section 2.1 Optics.

Figure 2: I think the dimensions of the measurement cell worth being noted in the figure.

Figure 2 has been updated and now shows the cell dimension.

L113: The data recording interval of sample detector and baseline detector is quite clear. The authors also performed dark signal check. How is this reflected in the data acquisition and analysis? Or is this just recorded for inspection?

The detector dark signal at the end of each scan is used to perform a zero-offset correction before spectrum normalization. This is now further explained inSection 2.3 Spectroscopy.

Figure 5: The abbreviation ADEV for Allan deviations needs to be explained while it is not note elsewhere.

The definition is now provided in Section 3.3 Noise.

Figure 6: Element 11 missing in the caption.

A temperature sensor was added to the Figure as element 11. The numbering in the caption was changed accordingly.

Figure 7 (bottom left and right), 8 (top), 9, 10 and 13: I find the units of $H_2O$ mixing ratio in "ppbv" are unnecessary because the lowest $H_2O$ volume mixing ratio to be detected was a few ppmv.

All plots were changed to units of ppmv.

L201: "is" -> in

Corrected.

L203: "where" -> were

Corrected.

Figure 9: In the first cycle, the prototype 1 obviously measured slightly lower values than the prototype 2 at each step above about 50 ppmv, which was not repeated in the second and third cycles. Do the authors have any ideas on the cause of this?

We were unfortunately not able to link this to a specific parameter or event. We rely on the statistical approach to intercompare the instruments for extended periods, as we did, to derive uncertainties.

References: 1 Not all DOI are inserted as links. 2 Not all journal names were abbreviated. 3 The style of the references with a long author list should be unified.

The reference style was adjusted to AMT standard.

**References mentioned in the comments:**

Buchholz, B., Afchine, A., and Ebert, V.: Rapid, optical measurement of the atmospheric pressure on a fast research aircraft using open-path TDLAS, Atmos. Meas. Tech., 7, 3653–3666, https://doi.org/10.5194/amt-7-3653-2014 , 2014.

Pogány A, Wagner S, Werhahn O, Ebert V. Development and Metrological Characterization of a Tunable Diode Laser Absorption Spectroscopy (TDLAS) Spectrometer for Simultaneous Absolute Measurement of Carbon Dioxide and Water Vapor. Applied Spectroscopy. 2015;69(2):257-268. doi:10.1366/14-07575

Buchholz, B. and Ebert, V.: Absolute, pressure-dependent validation of a calibration-free, airborne laser hygrometer transfer standard (SEALDH-II) from 5 to 1200 ppmv using a metrological humidity generator, Atmos. Meas. Tech., 11, 459–471, https://doi.org/10.5194/amt-11-459-2018, 2018.

May, R.D. (1998), Open-path, near-infrared tunable diode laser spectrometer for atmospheric measurements of H2O, J. Geophys. Res., 103, 19161-19172, doi:10.1029/98jd01678.

Kaufmann, S., Voigt, C., Heller, R., Jurkat-Witschas, T., Krämer, M., Rolf, C., Zöger, M., Giez, A., Buchholz, B., Ebert, V., Thornberry, T., and Schumann, U.: Intercomparison of midlatitude tropospheric and lower-stratospheric water vapor measurements and comparison to ECMWF humidity data, Atmos. Chem. Phys., 18, 16729–16745, https://doi.org/10.5194/acp-18-16729-2018, 2018.

Neis, P., Smit, H. G. J., Krämer, M., Spelten, N., and Petzold, A.: Evaluation of the MOZAIC Capacitive Hygrometer during the airborne field study CIRRUS-III, Atmos. Meas. Tech., 8, 1233–1243, https://doi.org/10.5194/amt-8-1233-2015, 2015.

Li, Y., Mahnke, C., Rohs, S., Bundke, U., Spelten, N., Dekoutsidis, G., Groß, S., Voigt, C., Schumann, U., Petzold, A., and Krämer, M.: Upper-tropospheric slightly ice-subsaturated regions: frequency of occurrence and statistical evidence for the appearance of contrail cirrus, Atmos. Chem. Phys., 23, 2251–2271, https://doi.org/10.5194/acp-23-2251-2023, 2023.

Unterstrasser, S. and Gierens, K.: Numerical simulations of contrail-to-cirrus transition – Part 1: An extensive parametric study, Atmos. Chem. Phys., 10, 2017–2036, https://doi.org/10.5194/acp-10-2017-2010 , 2010.

Petzold, A., Neis, P., Rütimann, M., Rohs, S., Berkes, F., Smit, H. G. J., Krämer, M., Spelten, N., Spichtinger, P., Nédélec, P., and Wahner, A.: Ice-supersaturated air masses in the northern mid-latitudes from regular in situ observations by passenger aircraft: vertical distribution, seasonality and tropospheric fingerprint, Atmos. Chem. Phys., 20, 8157–8179, https://doi.org/10.5194/acp-20-8157-2020, 2020.

Gierens, K.; Matthes, S.; Rohs, S. How Well Can Persistent Contrails Be Predicted? Aerospace 2020, 7, 169. https://doi.org/10.3390/aerospace7120169

---

## Author Comment (AC2)

We would like to thank both reviewers for the careful reading of the manuscript and the very constructive comments and suggestions, and we hope to have addressed them satisfactorily.

We have provided our answers to the comments and questions below in line in blue color.

**Review of "Accurate humidity probe for persistent aviation-contrail conditions" by Christoph Dyroff et al., MS No.: egusphere-2025-3972**

**Summary:**

This manuscript reports on a humidity sensor for the application of persistent aviation contrail conditions (i.e., humidity levels so high that contrails of condensed ice will remain persistent and not evaporate). It presents a novel optical design featuring optical fibers and a short-path absorption path. Two nearly identical prototypes are compared to each other (the only difference being electronics). The two prototypes show good agreement with each other over a large range of water volume mixing ratios.

**Overall:**

Overall, this manuscript is well-written and appropriate for the scope of AMT, presenting a new sensor prototype to measure humidity from aircraft. The title, abstract, presentation, use of language, and references are all good. The novelty of this sensor is both low-noise performance and optical design (optical fibers and a short-path absorption path). One issue is that the manuscript is lacking a connection between the scientific motivation of accurate humidity measurements at aircraft cruising altitudes and performance requirements for parameters including but not limited to: the desired range of water volume mixing ratio, pressures, temperatures, time resolution, accuracy and precision.

Another issue is that the description is not complete enough for subject matter experts to assess this new sensor. The manuscript is missing important details in the description of the instrument design and operation (the flow system), electronics, spectroscopic fitting, data processing, experimental validation, and how the results relate to the performance requirements. If the authors provided more details, then this would be an important contribution to the scientific literature. I recommend that this manuscript should be considered for publication only after substantial revisions to provide more details as addressed in the science comments below.

**General comments:**

1. First, early in the paper, the connection is not described between persistent contrails and the detailed range of water mixing ratios, pressures, temperatures expected. All that is said is that the "relevant H2O range below 200 ppm" is appropriate (page 15, line 274

and repeated elsewhere). This is missing detail such as: what is the lower limit of H2O mixing ratio expected in contrail-producing regions? What accuracy of measurement is required for this application of predicting persistent contrails? What is the spatial scale in the atmosphere of high humidity / low humidity regions that would drive a requirement of how fast measurements need to be? For instance, are 1-Hz measurements sufficient?

We have added a paragraph in the introduction to provide context for the expected humidity range for contrail conditions.

We have also provided context for the time response and linked spatial resolution.

The required accuracy is difficult to quantify other than better than the lowest expected humidity relevant to the question, here << 30 ppmv.

2. What are the performance requirements for sufficiently accurate water measurements at aircraft cruising altitudes? Specifically, what is the requirement measurement dynamic range of water volume mixing ratio, pressures, temperatures? What are the required time response, accuracy and precision?

See point 1 above.

3. Given that there are existing commercial instruments (WVSS-II and IAGSO ICH) and scientific-grade TILDAS hygrometers, what is the motivation for developing a new humidity probe? How is this probe novel? This not made very clear, but it is implied in the abstract that the novelty is single-mode optical fiber (and two channels). Could the authors please state why is the novelty important (compared to existing state-of-the-art sensors) for this application of measuring humidity in persistent contrail-forming conditions?

We have changed the wording from novel to new.

We have provided context for other commercially available technology and their benefits and limitations in the introduction.

4. For the readers, it is important to know hardware details, including material of the optical cell, and what electronics were used.

This information is now given in sections Optics and Electronics.

> 4a. In the lab measurements presented in this manuscript, what pressures and temperatures were used? Were these measured? Pressure and temperature are essential inputs into fitting spectra (as the water mixing ratio is dependent on them).
>
> This information is provided (160 Torr, 213 hPa, unstabilized room temperature).
>
> 4b. If these prototypes were deployed in an aircraft, how would design change? H would the air be sampled on an aircraft? How would the sensor be packaged?

What would be the pressure and temperature in the sample cell? How do gas control, temperature and pressure control affect the measurements?

We have added this information in the Outlook section, which has been added to the manuscript.

**Specific Comments:**

5. Spectroscopy: page 3, line 57-58: "an isolated absorption line of H2O". It is important to the readers to know specifically which H2O line? What is the wavelength of the isolated line? Can you show a synthetic spectrum of the expected absorbance versus wavenumber (or wavelength) (for the sensor pathlength)? Can you give some rationale for why this particular water absorption line was selected?

We have added Table 1 with parameters and their uncertainties in brackets if available from the HITRAN2016 database.

The fits in Figure 4 are essentially synthetic spectra. The figure now also indicates fractional absorption and noise metrics.

The rationales for choosing this line are provided in section Optics.

6. Detail is lacking on optical cell, such as the pathlength, how the detector is mounted. Specifically on page 3, line 65, "The beam is aligned through a pinhole target onto a detector at 300 mm." Is this (300 mm) the optical pathlength?

Figure 2 has been updated and now shows the cell dimension.

The text in section Optics was clarified to better describe the build process using a production jig with pinhole preceding the cell assembly.

7. In the Spectroscopy section, pages 4 and 5, can you provide more spectroscopic details, such as:

7a. In lines 88-89: please clarify how the baseline spectrum is used to normalize the sample spectrum. E.g., does this provide the incident/background intensity for the absorption calculation? Is it possible that trapped water in the laser diode or baseline detector could cause the low bias in the accuracy measurements (section 3.2)?

We have addressed this in more detail in the Optics (trapped water) and Spectroscopy section (fit).

7b. In lines 106-107: what is the motivation for the two different electronics in prototypes 1 and 2? Are there pros and cons of the different implementations?

We have clarified the choices in Section 2.2 Electronics.

We have used the NI system because it is a very matured system developed at ARI and provides the lowest-noise benchmark. The RedWave system was chosen for its compact size and integration of up to 3 low-noise laser drivers.

7c. In line 111, "2666 Hz (334 Hz)": why were these scan rates selected for prototypes 1 and 2?

This has been addressed in the Spectroscopy section.

7d. How many scans are averaged? Are scans fitted in the electronics or in software?

This is now better described in Section Spectroscopy.

7e. What is the sampling rate?

See 7c. for scan rate. The rate of $H_2O$ measurements was 1 Hz as described in 2.3 Spectroscopy.

7f. In line 114: how is 50 seconds of spectral averaging (followed by 10 s of averaging baseline) relevant to airborne measurements? Can the prototypes deliver 1-Hz data at the required accuracy and precision? (see comment #1 above)

The prototypes always deliver 1 Hz data. With prototype 1 we recorded 50 seconds of 1-sec sample spectra (not averaged) followed by a 10-second averaged baseline spectrum. The duty cycle was 50/60=83%. We have discussed the consequence of the duty cycle in the Outlook section.

7g. In line 103 "Upon pre-averaging of the spectra, they were fit in TDLWintel" and line 118 "we recorded spectra on the device and then fit them offline using the same fitting engine"– can you please say more about how the spectra are fitted? How many spectra are "preaveraged"?

This info is now provided in 2.3 Spectroscopy.

8. Page 5, lines 121-122: "glass rod" – please call this an etalon.

We have added that the glass rod "served as etalon" as it is not specifically designed to be an etalon.

9. Page 6, lines 130-134: discussion is completely lacking on how the spectroscopic absorption line is fitted. Can you please say more? Are you using the Beer-Lambert Law? How is pressure broadening treated?

We have extended the description in Section 2.3 Spectroscopy to include a better explanation of the treatment of sample and baseline spectra as well as the fit.

10. Page 6, line 133-134: "the noise is low enough to not significantly affect the overall measurement uncertainty" – this raises several questions:

This statement was vague and was removed.

We have moved the noise quantification to new section *3.2 Noise* and the following sections are re-numbered.

> 10a. What is the uncertainty of the spectral line strength (and other properties) for the water absorption line selected?

> See new Table 1. Linestrength uncertainty < 1%.

> 10b. If noise does not significantly affect the overall measurement uncertainty, what is the dominant factor contributing to overall measurement uncertainty?

> See above: This statement was vague and was removed.

> We have added a paragraph in the discussion regarding measurement uncertainty sources.

> 10c. Please provide more detail about how the electronics achieve exceptionally low noise? Please quantity the noise on the spectra.

> Figure 4 was updated. The noise in the baseline of the spectra is 1.8E-5 (1-sigma) in units of fractional absorption compared to a signal of 2.4E-4 (peak to peak) of the H2O line at 13 ppmv.

> We have added a discussion about how we achieved low noise in the new Section 3.2 Noise.

11. Page 7, Figure 5 caption: what are the details of this data? Which data went into making this plot? Can you please define the acronym ADEV (please say "Allan Variance").

We used the 1-sec data of Figure 7 (now Figure 6) as basis for the noise comparison. We used the Allan deviation because the deviation provides a more comprehensible unit (ppbv) than the variance. This is now defined in the text.

12. Figures 7, 8,10, 13: this manuscript jumps back and forth between ppm and "ppbv times 103". For better clarity, please consistently plot in the same units, ppmv.

We have changed all plots to units of ppmv.

13. Section 3.5 – how would the additional attenuation affect the accuracy and precision of the low-ppm measurement regime? It would also be valuable for the reader to have quantitative information on incident power limits for linear performance.

We have added a paragraph to Section Extended humidity range that discusses the noise increase due to attenuation.

We don't have enough statistics to quantify the incident power limits. The non-linearity was different for the two prototypes (two sets of detectors). This requires further experiments on a larger sample size to be quantitative.

14. Figure 3: Consider adding etalon peaks to subplot to show how the peak locations translate to wavenumber curve

We don't think it is necessary as the tuning rate curve is given.

15. Figure 7 and Figure 9 show the quick response time of the instrument in lab setting with the test configurations. Could you comment on the cadence / delay times expected when integrated with an aircraft and the associated sampling system?

We have added the response time of our prototypes to Figure 7 (now Figure 6). It was derived from a double-exponential fit to the $H_2O$ measurement on the falling edge from 50 ppmv to 0 ppmv at 1 slpm flow rate.

We have added a new Section 3.2 Response to humidity changes and included a brief paragraph regarding time response in the outlook section.

16. Will the sensors still be non-temperature controlled when integrated on aircraft? How will the range of temperatures and rapid fluctuations in temperature affect the instrument hardware and spectroscopic fitting?

We have added more specifics of the anticipated pre-production instrument in a new Section Outlook.

17. Section 3.5: Please elaborate on how the non-linearities affect the accuracy and the detection limits mentioned.

See point 13.

18. Page 15, Line 278-280: Please provide details of all items listed: "spectroscopic fit, including the laser-dependent tuning rate, the baseline characteristics as well as position and width of the fit." How do these contribute to the accuracy?

This has been addressed in the discussion.

**Editing Comments:**

1. Please check grammar. Several commas are missing where needed.

2. Consistently use ppmv (not ppm or ppbv) and define ppmv the first time as "parts per million by volume".

Done.

3. Page 10, line 201: replace "is" with "in".

Done.

4. References: please cite references in consistent EGU format.

Done.